# A value-based model of job performance

Michael Roos, Jessica Reale ◉ *, Frederik Banning ◉

Chair of Macroeconomics, Ruhr-Universität Bochum, Bochum, Germany

* jessica.reale@ruhr-uni-bochum.de

## Abstract

This agent-based model contributes to a theory of corporate culture in which company performance and employees' behaviour result from the interaction between financial incentives, motivational factors and endogenous social norms. Employees' personal values are the main drivers of behaviour. They shape agents' decisions about how much of their working time to devote to individual tasks, cooperative, and shirking activities. The model incorporates two aspects of the management style, analysed both in isolation and combination: (i) monitoring efforts affecting intrinsic motivation, i.e. the company is either trusting or controlling, and (ii) remuneration schemes affecting extrinsic motivation, i.e. individual or group rewards. The simulations show that financial incentives can (i) lead to inefficient levels of cooperation, and (ii) reinforce value-driven behaviours, amplified by emergent social norms. The company achieves the highest output with a flat wage and a trusting management. Employees that value self-direction highly are pivotal, since they are strongly (de-)motivated by the management style.

## 1 Introduction

Keeping a company's employees motivated to work on their assigned tasks is arguably one of the key goals of management. Economists traditionally assume that financial incentives are effective in exerting a desired influence on people's behaviour. There is an extensive literature on the effects and the design of rewards schemes at the workplace [1]. While economists tend to believe that performance-related rewards typically improve employees' performance [2], researchers with a psychological background are more sceptical and believe that financial incentives can backfire, for instance if they conflict with employees' desire for autonomy [3]. Psychologists argue that in many situations people are intrinsically motivated and that financial incentives can have adverse effects on desired behaviours, by crowding out internal motivation [3–5]. However, the dichotomy of incentives vs. intrinsic motivation is probably a too simplistic view on behaviour in the workplace, since both factors refer to individual agents in isolation. For a complete understanding of employee behaviour (and human behaviour in general), social interactions should be taken into account. Organisational culture is another important element impacting employees' efforts and performance [6]. Employees follow social norms about appropriate behaviour that operate through informational and normative social influence [7]. So far, the interaction between monetary incentives, intrinsic motivation and social norms and their joint effect on employee behaviour has received little attention.

This paper presents a theoretical agent-based model that treats monetary incentives, motivational factors and endogenous social norms as joint determinants of employees' work effort and the resulting company performance. We use the model to answer the question of how

**Data Availability Statement:** The complete codebase to recreate the data is available at a public repository here: https://git.noc.ruhr-uni-bochum.de/vepabm/invano-public; The same information is included in the manuscript in section 2.4 "Method".

**Funding:** This study was supported by Deutsche Forschungsgemeinschaft (DFG) (Project number 426603947) awarded to MR, and the Open Access Publication Funds of the Ruhr-Universität Bochum. The funders had no role in study design, data collection and analysis, decision to publish, or preparation of the manuscript.

**Competing interests:** The authors have declared that no competing interests exist.

incentives set by different remuneration systems affect shirking and cooperative behaviour in different organisational cultures. In particular, we compare the effects of a uniform payment scheme with identical salaries for all employees, an individual reward scheme with personal performance-related incentives and a collective reward scheme, in which incentives depend on group performance. A company's management can partly influence the organisational culture, e.g. by monitoring employees, but culture also evolves endogenously by employees' perception of what constitutes normal behaviour. The paper fills a gap in the research literature, since the influence of corporate culture and remuneration are usually treated in separate strands of the literature. Chatman et al. [7] call for the development of a theory of corporate culture that explains how culture affects company performance and how it interacts with elements of organisational structure such as the remuneration system. Huck et al. [8] present a model with homogeneous utility-maximising agents in which there is an interplay between economic incentives and social norms in companies. Our model contributes to such a theory in a different way. In contrast to Huck et al. [8], we do not assume that all workers are identical, but that they differ in their personal values which leads to different behaviours. Since an equilibrium model with heterogeneous agents would be intractable, we use an agent-based model. We use our model to answer the research question: What is the impact of organisational culture and remuneration schemes on corporate performance when employees are heterogeneous in their personal values?

The starting point of our analysis is the assumption that human behaviour is guided by social norms to a considerable extent. Elsenbroich et al. [9] argue that in fact most human behaviour is governed by social, moral or legal norms. They define social norms as rules of conduct derived from social behavioural expectations. The source of moral rules are moral values such as honesty, fairness, respect or responsibility. Since Deutsch et al. [10], it is common to distinguish descriptive norms, exerting informational social influence, and injunctive norms, having normative social influence. While injunctive norms express the (subjective) expectations of others regarding one's own behaviour, descriptive norms describe the "normal" behaviour of the group, i.e. what is actually done on average. We use the concept of descriptive norms and model it as the observed average behaviour of other employees. Since norms depend on actual behaviour, they can change over time such that a part of the corporate culture is endogenous.

A theory of corporate behaviour and performance based on culture, external incentives and intrinsic motivation must allow individuals to deviate from the social norms and explain why and how this happens. We postulate that the responsiveness to external incentives and the compliance with social norms depend on the individuals' values. In particular, we assume that employees differ in their values according to the well-established theory of personal values of Schwartz et al. [11–14]. According to Schwartz et al. [11], basic values are trans-situational goals, varying in importance, that serve as guiding principles in the life of a person or group. Since the theory postulates that individuals prioritise the values differently, we assume that agents can be classified into four types: an ST-type (self-transcendent), an O-type (open to change), an SE-type (self-enhancing) and a C-type (conserving). These types differ in how they respond to monetary incentives, how much their behaviour is self- or norm-guided and how cooperative they are.

Company performance of course depends not only on the individual effort of each employee, but also on how much employees cooperate with each other. While the degree to which the output of a company depends on cooperation differs across industries, there are very few production processes that do not require some direct collaboration of co-workers. From the perspective of a company's management, everyone in a work environment is expected to dutifully accomplish their assigned tasks within their working hours. In reality, however, there is little doubt that people often prefer leisure over work activities, i.e. the extent of mustered effort will settle on a submaximum level [15] with up to 86% of employees self-

reporting shirking behaviour in a recent survey [16]. Shirking at work covers the active decision to be unproductive [17] and can manifest itself in a variety of forms like checking personal emails and social media or prolonged chatting with coworkers about topics unrelated to work. Furthermore, employees may not be willing to collaborate sufficiently with their colleagues, either because they do not like working together with others or because they believe to have a strategic career advantage if they work on their own. Companies hence want to influence their employees' behaviour such that they do not shirk and collaborate sufficiently.

One way to reduce shirking is to give employees precise instructions for their behaviour and to monitor their actions closely. In our model, we describe this as a *controlling environment* in contrast to a *trusting environment*, in which the management does not prescribe in detail how the employees are expected to perform their work assignments. The other conventional instrument of preventing shirking and fostering collaboration is to install performance-based remuneration schemes with either individual or group rewards. While both types of schemes are expected to reduce shirking, the degree of collaboration might depend on whether rewards are paid for individual or group performance.

The outline of this paper is as follows. Section 2 presents the model, i.e. the methodology, the environment and agents' behavioural rules. Section 3 explains the simulations and the main results of the model and the last section concludes.

## 2 Materials & methods

Our theory relies strongly on the assumption that agents are heterogeneous with regard to their most important values. Furthermore, there is a feedback mechanism between agents' behaviour and social norms, which leads to a co-evolution of norms and behaviour over time. For these reasons, we develop an agent-based model which is the best way to incorporate heterogeneity and to analyse these co-evolutionary dynamics. Fig 1 presents the overview of the model.

### 2.1 Production technology

There are N employees in the company. For simplicity, they are all identical in terms of their productivity and skills, given the abstract nature of the production process we aim at describing, but differ in terms of their values as discussed in the next subsection. Every employee $i$ has a daily time budget $\tau$ that can be allocated to three distinct activities: working on individual tasks $p_{i,t}$, collaborating with others $c_{i,t}$ and shirking $s_{i,t}$, resulting in the following time restriction for all employees: $\tau = p_{i,t} + c_{i,t} + s_{i,t}$. All variables are updated each period $t$, which corresponds to a working day. The output of employee $i$ is given by a function of the Cobb-Douglas type and depends on the time devoted to the employee's own work, $p_{i,t}$, and the average time other team members devote to cooperation, $\bar{c}_{i,t}$, defined in Eq 2.

$$O_{i,t} = p_{i,t}^{1-\kappa} * \bar{c}_{i,t}^{\kappa} \tag{1}$$

Eq 1 can be interpreted as an individual production function that relates the output of a single employee to the labour input. The parameter $\kappa$ measures the degree to which the output of employees depends on the support of their co-workers, which is called task interdependence [18]. In principle, $\kappa$ could depend on the specific job characteristics of each employee within the company. For simplicity, we assume that all jobs are similar such that $\kappa$ is identical for all agents. Task interdependence generates a need for collaboration. The average time of the co-workers of $i$ devoted to collaboration is defined in Eq 2.

$$\bar{c}_{i,t} = \frac{1}{(N-1)} \sum_{j \neq i} c_{j,t} \tag{2}$$

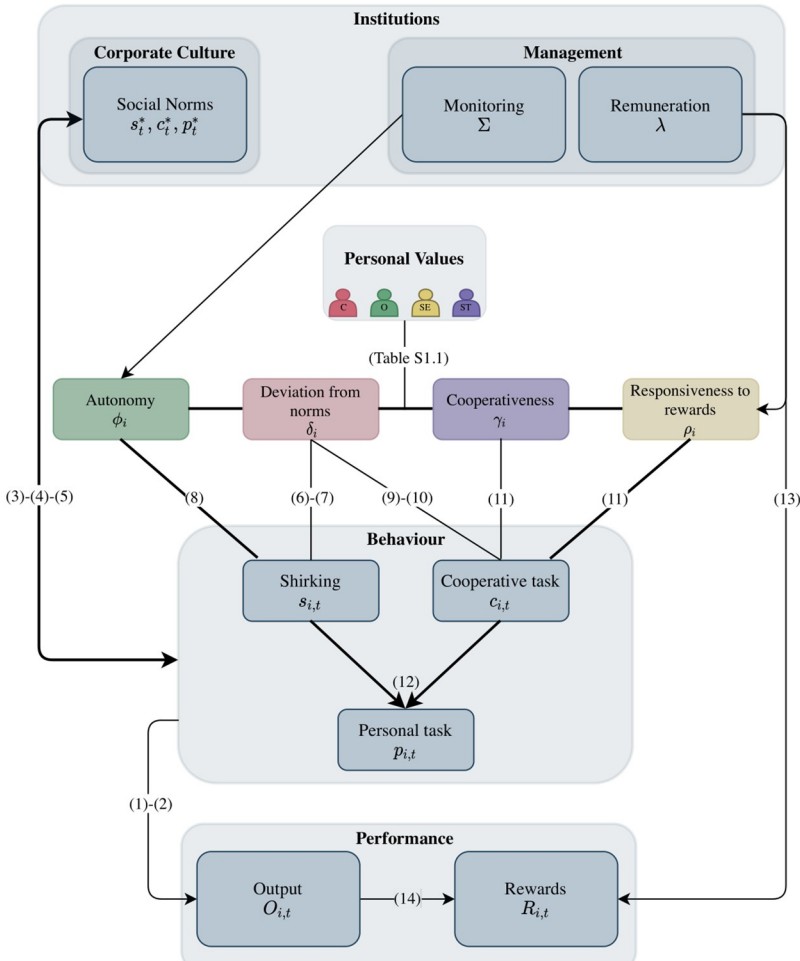

**Fig 1. Model overview.** The numbers in parenthesis are equation numbers.

Time that an employee spends on shirking $s_{i,t}$ is not productive, therefore it does not contribute to output (see Eq 1).

## 2.2 Agent behaviour based on social norms

A key assumption of our model is that agents' behaviour is anchored by the social norms of the relevant peer group, which is the company's workforce in our case. We do not want to use a utility-maximising framework, because we consider such a setting as highly unrealistic for the behaviour we want to model. Behavioural research suggests that human thinking can be described by two different modes [19]. *System-1 thinking* is fast, effortless, emotional and unconscious. In contrast, *system-2 thinking*, which corresponds to rational maximisation behaviour, is slow, requires active effort and must be activated consciously. We argue that the organisation of daily activities is mostly guided by system-1 thinking. Since system-2 thinking requires effort and cognitive resources are limited, humans economise on the use of cognitive resources and apply system-2 thinking only when it pays off doing so. Following social norms of behaviour is an efficient way of organising our daily life. This reasoning suggests that agents reserve system-2 thinking for their job tasks, but use system-1 thinking for the allocation of

their daily working time. As a consequence, we assume that the time allocation is not determined by solving an effortful utility-maximisation problem, but by an effortless interplay between following social norms and ad-hoc deviations from these norms driven by contextual and affective factors.

For every possible activity (i.e. production, cooperation or shirking), there is a norm that reflects what is seen as normal in the organisation. Hence there is a norm for accepted shirking behaviour, $s_t^*$ like chatting with colleagues, sending private emails or smoking cigarettes during working time.

$$s_t^* = (1 - h) \, s_{t-1}^* + h \, \frac{\sum_{j \in N} s_{j,t-1}}{N} \tag{3}$$

There is also a norm for helping others and participating in group activities, $c_t^*$.

$$c_t^* = (1 - h) \, c_{t-1}^* + h \, \frac{\sum_{j \in N} c_{j,t-1}}{N} \tag{4}$$

We assume that behaviour is driven by descriptive norms, which means that the social norms are a weighted average of the prevailing norms in the previous period, $c_{t-1}^*$ and $s_{t-1}^*$, and the average of all agents' behaviour in the previous period.

From these norms, the social norm for individual working time follows from the time constraint, see Eq 5.

$$p_t^* = \tau - s_t^* - c_t^* \tag{5}$$

In the initial period of the simulation, all agents start with identical exogenously given time allocations. In the subsequent periods, the social norms are updated by the most recent observed behaviour which is weighted with a constant factor $h$, ranging from 0 (constant social norms) to 1 (fully adaptive social norms). The updating factor measures the persistence of social norms or the stability of the corporate culture. We set $h = 0.1$ resulting in a rather slow evolution of the norms. The impact of varying values for $h$ has been tested in the sensitivity analysis described in S3 Section in S1 File.

We have tested two additional local environments for social norms. However, the results of these simulations converged towards the global norm outcomes. For this reason, these supplementary specifications along with the similarity between the three chosen environments are provided in S2 Section in S1 File. For the sake of generalisation, we use an asterisk to denote social norms independently of their scope. Since social norms coincide with the average time spent on any activity across the whole population in a *Global* scope of social norms, it might be possible to adopt the usual notation for averages (i.e. $\bar{c}_t$ or $\bar{s}_t$). However, this would not be valid for the two additionally tested local scopes (see S2 Section in S1 File). Hence our use of the asterisk which avoids this potential source of notational inconsistency.

Social norms guide agents' behaviour, but they do not fully determine it. The actual behaviour of each agent on a working day is influenced by a host of other factors, such as personal mood, fatigue, pressure by deadlines, attractiveness of the required tasks, team spirit, support by colleagues and superiors. We capture all of these affective and contextual factors by stochastic deviations from the norms. System 1 uses affective and contextual cues in order to adjust behaviour away from the norm, if the situation requires this.

Every day, employee $i$ might deviate from the shirking $s_t^*$ and the cooperative norm $c_t^*$. This deviation can be positive, negative or zero.

For example, employees might shirk more than normal, if they have to do a boring task or are in a depressed mood. Less shirking might occur, if a task is perceived as interesting or the

team spirit is motivating. The employee might be less inclined to cooperate on a given day, if there was an argument in the team. More cooperation might happen on sunny days, when everybody is in an elated mood.

To model the stochastic deviations from the norms we use a triangular distribution ($T$), which has the convenient property of being bounded between two parameters, $a$ and $b$. The actual time employees spend on a generic activity at a given time $x_t$ is derived as a random draw from the triangular distribution, that is $x_t \sim T(a, b, m)$ where $m$ is the mode of the distribution, see Fig 2.

The natural lower bound is $a = m - m = 0$ because employees cannot allocate negative amounts of time to an activity. For simplicity, we assume that the upper bound is $b = m + m$, which means that an employee will never spend more than twice the time on an activity compared to what is currently seen as social norm. The upper bound could be interpreted as a management or leadership parameter, as it might depend on what is accepted by the management of the company. For the moment, we assume that the distribution is symmetric, hence the mode of the distribution is equal to the social norm $m$ which is also the mean since $E(x_t) = {(a+b+m)}/{3} = {3m}/{3} = m$. In this baseline case, employees would most likely spend their time according to the norm. Employees might also shirk/cooperate more or less but with decreasing probabilities of larger deviations. We assume that agents differ intrinsically regarding their willingness to comply with social norms and to cooperate. The underlying reason for this difference is the heterogeneity in agents' value priorities.

## 2.3 The effect of values on behaviour

The Schwartz theory of basic values is a well-established theory that has been empirically assessed with data from hundreds of samples in 82 countries of the world [20]. Originally, Schwartz identified 10 basic values, which were later extended to 19 values [21]. The Schwartz theory says that those values are organised in a coherent system that underlies individual

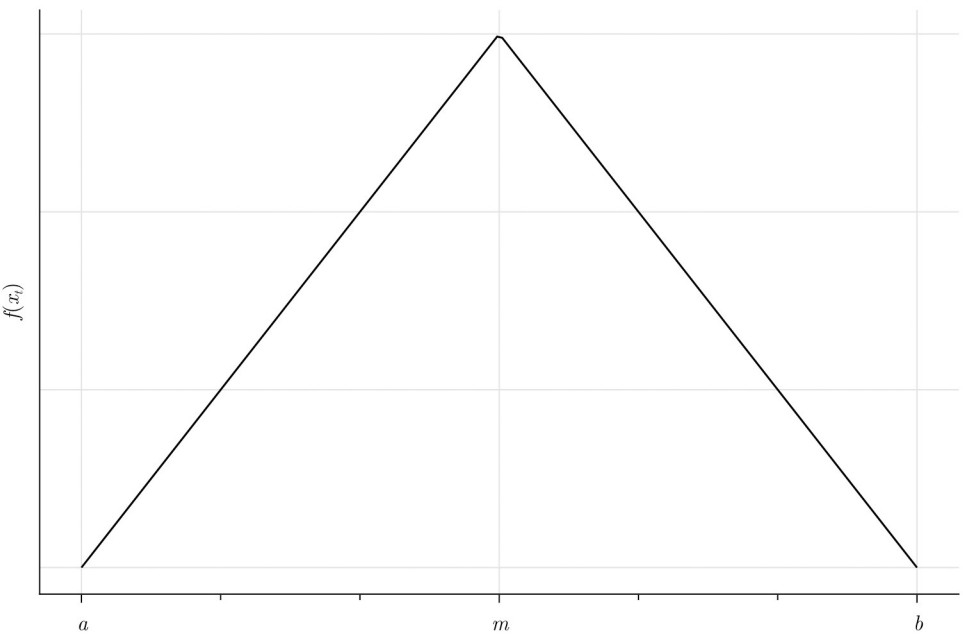

**Fig 2. Density function of the triangular distribution for a generic social norm.**

decision making. In this system, the values can be arranged on a circle, with neighbouring values being similar and having similar effects on behaviour, while values that are further apart on the circle are more different (see Fig 3). A second important element of the theory is that individuals have a hierarchy of values with some values being more important to them than others.

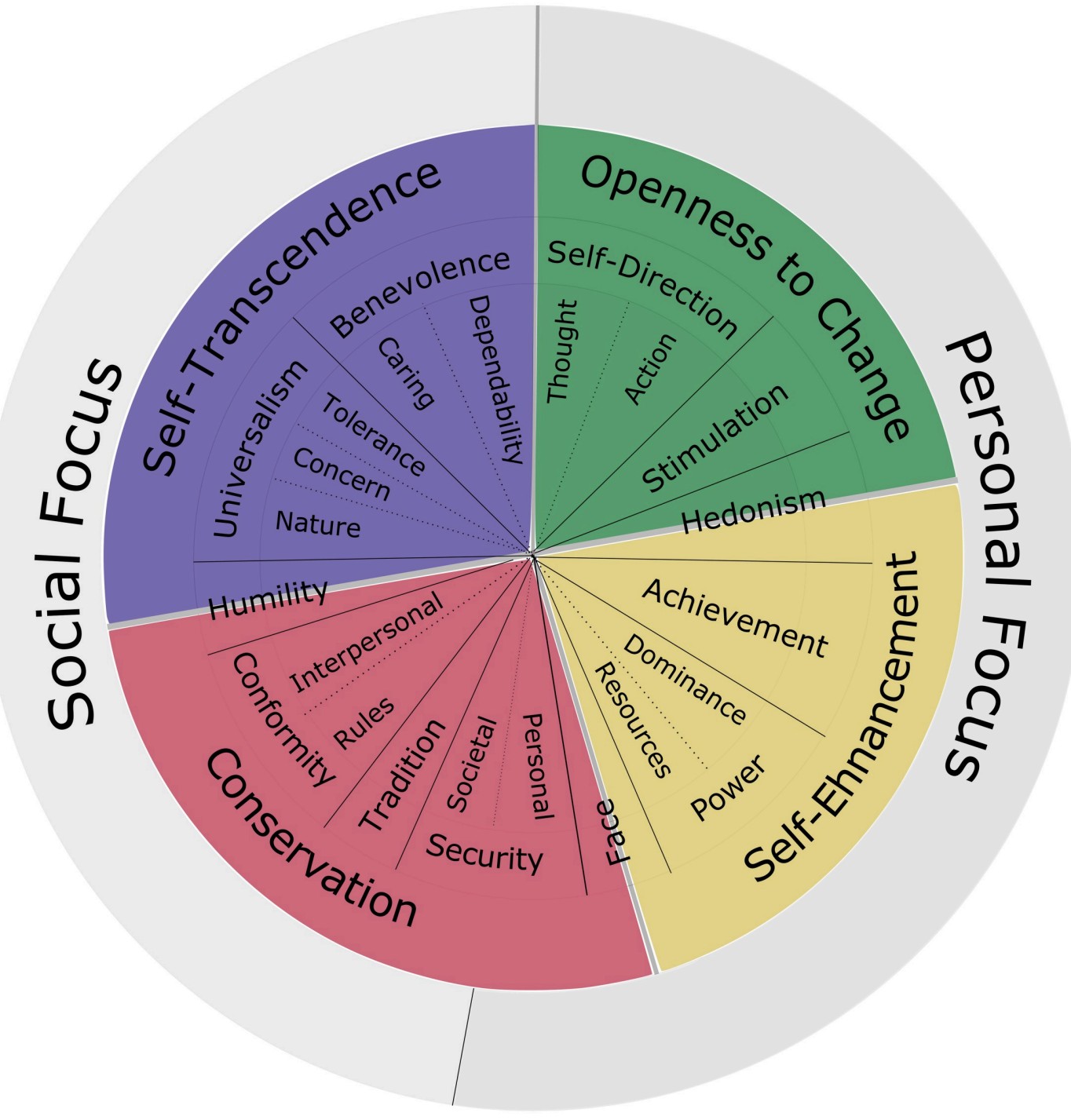

**Fig 3. Circular motivational continuum source: Adapted from Schwartz et al. [21].**

Factor analyses show that the basic values can be aggregated along two dimensions [22]. The first dimension contrasts aspects of self-transcendence and self-enhancement. Self-transcendence comprises the values of universalism and benevolence which are oriented towards the well-being of others. In contrast, self-enhancement is focused on power and achievement which aim at personal well-being. The second dimension is about openness to change and conservation. Basic values related to openness are self-direction and stimulation, which motivate behaviours that aim at experiencing freedom, excitement, novelty and change. The values connected to conservation are conformity, security, preservation of traditions and stability.

In line with the theory, we assume that agents have a value hierarchy and can hence be categorised into types according to their most important values. For simplicity, we consider four types: ST-agents driven by the self-transcendence values (e.g. benevolence), SE-agents motivated by power and achievement, C-agents for which the conservation values security and conformity are most important, and O-agents for whom self-direction (openness to change) is the main motivator. Agents' heterogeneity in terms of their most important values impacts their behaviours along four attributes: (i) deviation from social norms; (ii) inclination to cooperative behaviour; (iii) sense of self-direction or autonomy, relevant for intrinsic motivation; (iv) responsiveness to financial rewards.

The Schwartz theory does not only assume a hierarchy of values, but also a circular structure. This implies that types of agents at opposite ends of the openness-to change vs. conservation dimension and the self-transcendence/self-enhancement dimension are most different in their behaviour. Neighbouring types are more similar. To have a clearer understanding of the dynamics of the model, we do not assume the existence of mixed types arising from the combination of the two aforementioned dimensions (ST-O, ST-C, SE-O, SE-C). For the purpose of this paper, it is convenient to isolate the impact of each value dimension separately. Table 1 shows how we map the value hierarchy and the circular structure on the four attributes of employees' behaviour.

We assume that the openness-to-change vs. conservation dimension (i) is relevant for agents' tendency to deviate from social norms and their need for autonomy and (ii) affects employees' shirking behaviour ($s_{i,t}$).

The self-transcendence/self-enhancement dimension determines cooperativeness and responsiveness to rewards, influencing cooperative decisions ($c_{i,t}$) in our model.

The amount of time spent on shirking is derived from a triangular distribution with parameters as defined in Eqs 6–8.

$$a_{i,t} = s_t^*(1 - \delta_i) \tag{6}$$

$$b_{i,t} = s_t^*(1 + \delta_i) \tag{7}$$

$$m_{i,t} = s_t^* + \phi_i \tag{8}$$

**Table 1. Value types and attributes.**

| Type | Deviation from Norms ($\delta$) | Cooperativeness ($\gamma$) | Autonomy ($\phi$) | | Responsiveness to Rewards ($\rho$) | |
|------|------|------|------|------|------|------|
| | | | Trusting | Controlling | Individual | Group |
| C | low | medium | low | **high** | medium | medium |
| O | **high** | medium | **high** | low | medium | medium |
| SE | medium | low | medium | medium | **high** | low |
| ST | medium | **high** | medium | medium | low | **high** |

Analogously, the triangular distribution parameters determining time spent on cooperation are defined in Eqs 9–11.

$$a_{i,t} = c_t^*(1 - \delta_i) \tag{9}$$

$$b_{i,t} = c_t^*(1 + \delta_i) \tag{10}$$

$$m_{i,t} = c_t^* + \gamma_i + \rho_i \tag{11}$$

The actual time an employee spends on individual tasks is defined residually after $s_{i,t}$ and $c_{i,t}$ are randomly drawn from the triangular distribution, as in Eq 12.

$$p_{i,t} = \tau - s_{i,t} - c_{i,t} \tag{12}$$

The following subsections describe how we model these assumptions and how they relate to agents' behaviour.

**2.3.1 Deviation from norms ($\delta$).** The scaling factor $\delta \in [0, 1]$ scales down the maximum of the lower and the upper bound such that the greater the importance an agent-type gives to social norms, the higher is the probability that the employee will follow the norm and the smaller will be potential deviations. We use the parameterisation shown in Table 2: C-agents (O-agents) have the lowest (highest) probability to deviate from the social norms, while SE- and ST-agents have intermediate probabilities, as shown in Fig 4.

**2.3.2 Cooperativeness ($\gamma$).** The parameter $\gamma \in [-1, 1]$ shifts the mode of the distribution to the left or to the right on the basis of agents' natural willingness to *cooperate with others*. Some types are likely to cooperate more than the norm, others less, which we model by the skewed density functions shown in Fig 5. The violet function of the ST-type is left skewed placing more probability mass on positive deviations from the cooperation norm ($c_t^*$). Hence, ST-agents on average cooperate more than the norm and the other types, i.e. $\gamma > 0$. The opposite holds for the SE-type, which has the yellow right skewed density function ($\gamma < 0$). The green function of the O- and the magenta one of the C-type agents are the intermediate cases which are centered around the cooperation norm.

**2.3.3 Autonomy ($\phi$).** Value types differ also in the attribute *need for autonomy*. Autonomous motivation of agents can stem from various sources like identified regulation (from trust and reciprocity) and intrinsic motivation (from needs, values, knowledge, cohesiveness). In this version of our model we leave out the former and focus on intrinsic motivation which is specifically governed by personal values. Note that there is an important distinction between the personal need for autonomy, as just one activating factor in terms of intrinsic motivation, and the autonomous motivation itself which can be understood as a supercategory covering its subcategory intrinsic motivation. For the rest of this paper we will use the term "autonomy" to refer to agents' personal (need for) autonomy and "autonomous motivation" to refer to aspects of intrinsic motivation.

**Table 2. Probability to deviate from norms.**

| Types | Deviation from Norms | $\delta$ |
|-------|----------------------|----------|
| C | low | 13 |
| O | **high** | 1 |
| SE | medium | 23 |
| ST | medium | 23 |

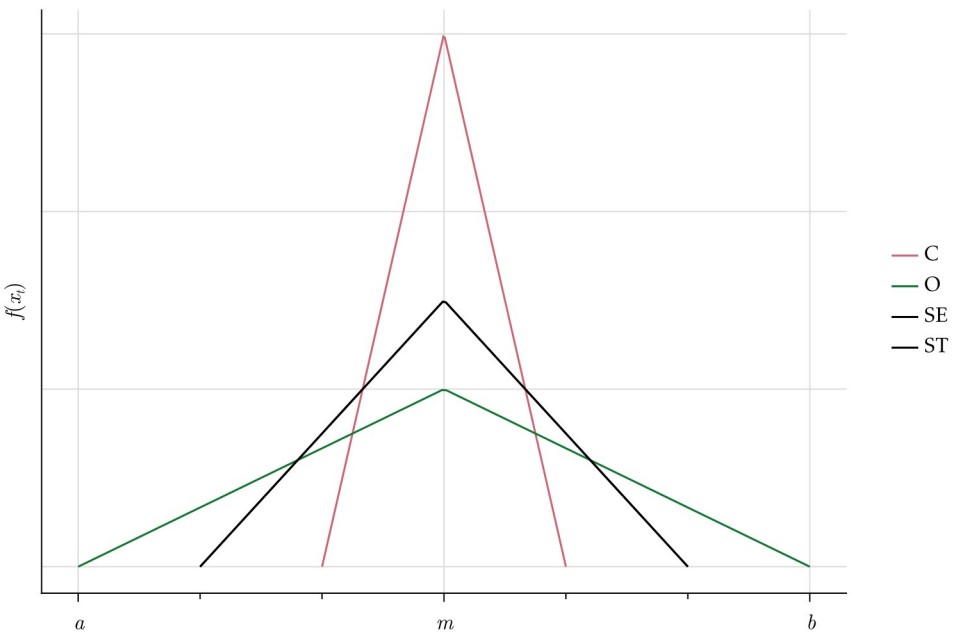

**Fig 4. Deviation from norms.**

Self-determination theory [23] argues that the feeling of autonomy and intrinsic motivation are linked. Since the value of self-direction is part of the openness dimension, we assume that intrinsic motivation is more important than extrinsic motivation for O-type agents. In contrast, C-type employees value security more than self-direction, which implies that their job motivation is more extrinsic than intrinsic. In other words, C-employees work for money in

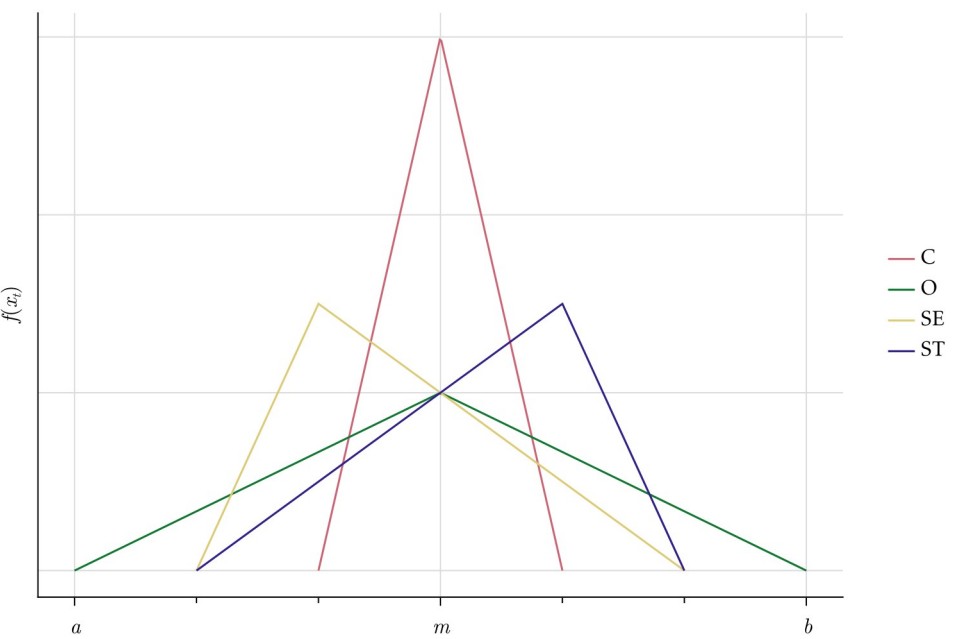

**Fig 5. Density functions for cooperativeness.**

order to satisfy security needs, whereas the job can be a means to experience autonomy for O-employees.

We assume that autonomy and intrinsic motivation are relevant for agents' shirking behaviour. It is a well-documented finding that employees shirk more or perform other counterproductive work behaviour when their job satisfaction is low [24–26]. Job satisfaction and work motivation, in turn, are likely to be influenced by the management style which is part of the organisational culture. In this regard, the management style can matter in two respects. First, job satisfaction and hence workplace deviance depend on employees' perception of being treated fairly [26]. Second, whether employees experience autonomy also depends on the management style. In our stylised model, we conceive the management style and hence the corporate culture as dyadic, meaning that it can either be trusting or controlling. In a trusting culture, employees are granted the freedom to organise their work as they like. In contrast, in a controlling culture, employees are constantly monitored by their superiors and receive detailed instructions about what to do and what to omit.

How employees respond to the management style depends on their value type. We assume that employees of the SE-type and the ST-type are relatively unresponsive to whether the organisational culture is trusting or controlling. However, the O-type and the C-type respond in opposite ways. The O-type employees flourish in a trusting culture because they value freedom and experience autonomy. They hence tend to work more and shirk less than the norm. In contrast, the C-type employees feel insecure by the absence of clear rules and instructions. They might interpret their freedom as disinterest of the employer which demotivates them or induces the belief that they do not have to work hard. In this interpretation, they reciprocate perceived disinterest with low effort which appears morally justified. In a controlling culture, the situation is exactly the opposite.

We control the skewness of the density functions with a parameter $\phi \epsilon [-1, 1]$. This parameter depends on whether the employee is motivated by the organisational culture to shirk more or less than the norm.

Hence, $\phi$ can be interpreted as the degree of frustration with the management style. If $\phi > 0$, agents' need for autonomy conflicts with the management style, resulting in a higher probability to shirk more than the norm. Fig 6 shows how these considerations are modelled in terms of the density function of the deviations from the shirking norm ($s_t^*$).

**2.3.4 Responsiveness to rewards ($\rho$).** Finally, values also affect how employees respond to financial rewards. We compare three different remuneration systems. In the baseline case, all agents receive the same fixed wage base, $\omega_b$, independently of their output, which is equal to an hourly wage $\omega$ (set equal to one) times the daily working time $\tau$.

Financial rewards or bonuses are a classic instrument companies use in order set incentives for employees to increase their work effort. Bonus-based plans are versions of the so-called

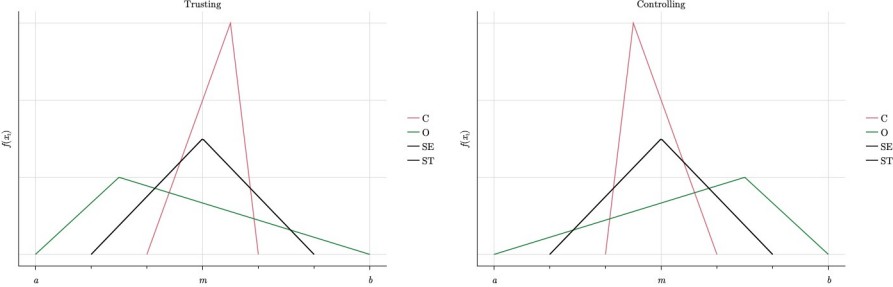

**Fig 6. Deviations from shirking norm in different organisational cultures.**

pay-for-performance (PFP) plans and can be a function of individual output and (average) group output. Following Wageman and Baker [27], the bonus $B_{i,t}$ paid to employee $i$ can be expressed as in Eq 13.

$$B_{i,t} = (1 - \lambda)O_{i,t} + \lambda \left(\frac{1}{N}\right)\sum_{j=1}^{N} O_{j,t} \qquad (13)$$

The parameter $\lambda \, \epsilon \, [0, 1]$ measures the degree of reward interdependence. When $\lambda = 0$, employees receive bonuses only according to their own output, such that there is no reward interdependence. On the contrary, reward interdependence is maximised for $\lambda = 1$, when agents are paid for joint production only, formalised as the average sum of all employees' output ($N$). For the sake of simplicity, we focus on these extreme cases and do not take into account the intermediate case of mixed PFP schemes with $0 < \lambda < 1$. We call the case with $\lambda = 0$ a *competitive* reward scheme because it sets an incentive for individuals to maximise their individual output. The other case with $\lambda = 1$ is called *cooperative* reward scheme. The total reward of an employee $i$ is the sum of the base wage plus the bonus, if the company uses a PFP plan, expressed by the indicator $\mu = 0 \vee \mu = 1$.

$$R_{i,t} = \omega_b + \mu B_{i,t} \qquad (14)$$

Regarding the question of how agents respond to financial rewards, our approach differs most clearly from a conventional utility-maximisation approach. In a utility-maximisation framework, one would assume that employees have a taste for money and a distaste for effort. Such a framework, however, has problems of incorporating intrinsic motivation and social effects.

The conventional assumption is that workers must be paid as a compensation for their disutility from working. With utility-maximising agents, it would be natural to assume that all agents choose the maximum level of shirking, if the financial remuneration is unrelated to effort ($\mu = 0$). This maximum level of shirking might be derived from expected costs and benefits of shirking, which depend on the likelihood of getting caught and punished and the expected value of the punishment.

Introducing a performance-related remuneration element ($\mu = 1$) would reduce the optimal level of shirking, because the monetary cost of shirking goes up. While we acknowledge that there might be some effects of bonuses on shirking behaviour, we argue that finding the utility-maximising level of shirking is a rather difficult or even intractable optimisation problem. We assume here that the dominant effects on shirking are related to intrinsic motivation and the interaction between the need for autonomy and management style (as described in Section 2.3.3) and not to financial bonuses. Along these lines, Nosenzo et al. [28] provide strong experimental evidence that financial bonuses do not reduce shirking in an inspection game.

We assume that PFP schemes affect employees' willingness to cooperate. According to goal-framing theory [29], we can distinguish individual and supra-individual mindsets of employees. A goal frame activates a specific overarching goal in one of these mindsets by making it focal. The *normative goal frame* emphasises a collective "We-orientation", i.e. one of collective goals and collaboration. In contrast, the *gain goal frame* puts the focus on an individual's personal self and private material gains, fostering an "I-orientation". In line with goal-framing theory, we assume that the type of the PFP scheme provides a goal frame that either strengthens a We-orientation or an I-orientation. Paying individual rewards, the company provides a gain frame which promotes individual effort and hinders cooperation. Group rewards, in turn, communicate to employees that they have a common goal which can be achieved better by collaboration. The company hence sets a normative goals frame with group rewards.

Burks et al. [30] show in an artificial field experiment with bicycle messengers that individual performance pay in fact reduces cooperation. Lee et al. [31] confirm goal-framing theory with data from a natural experiment in South Korea. Their findings support the existence of social effects of rewards schemes on cooperative behaviour which are in line with goal-frame theory but contradict the alternatives of agency theory and equity theory. In particular, they find that cooperation increases after a switch from an individual PFP scheme to fixed pay. Using a sequential prisoners' dilemma game, Burks et al. [30] measure the cooperative predispositions of their experimental subjects and categorise them into egoists, altruists and conditional cooperators. Whereas egoists always defect, altruists always cooperate, regardless of what the first-mover has done. The experiment shows that individual performance pay appears to strengthen preexisting egoism, which means that the effect of the reward scheme is especially large on egoists. We translate egoists with SE-type agents in the Schwartz terminology and altruists with ST-type agents. Following our previous reasoning, we assume that C-agents and O-agents are between the other types and hence do not show a significant tendency towards more or less cooperation in response to rewards schemes.

Since Burks et al. [30] only report a strong negative effect of individual performance pay on egoists' cooperativeness, we assume that individual rewards in a competitive rewards scheme make SE-agents cooperate much less. As shown in the right panel of Fig 7, this incentive effect is less pronounced for ST-agents because they are generally adverse to reducing cooperative efforts. By analogy, we assume the opposite effects of group-based rewards in a cooperative rewards scheme: ST-agents become significantly more cooperative, whereas SE-agents become only slightly more cooperative because they still react to incentive structures even though they don't fully match their preferences.

The competitive rewards scheme emphasises individual output and achievement and hence is well-aligned with the most important goals of the SE-agents, but in conflict with the prime goals of the ST-agents. ST-agent still might cooperate slightly more because more cooperation increases the output of others if there is task interdependence. A similar reasoning applies in the opposite case of the cooperative reward scheme, under which an individual effort indirectly also pays off since it also contributes to the average group output. These assumptions are formalised via the parameter $\rho \epsilon [-1, 1]$, following the same logic explained in the previous paragraphs.

## 2.4 Method

Our agent-based model is implemented in the Julia Programming Language [32, see also https://julialang.org/] and makes use of multiple packages from the Julia ecosystem [33–37]. The complete codebase has been made available online (here), including the required Julia

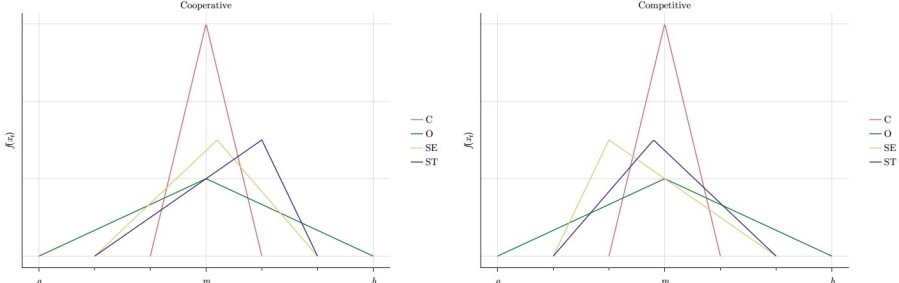

**Fig 7. Cooperative vs. competitive reward scheme.**

environment, the model itself and auxiliary scripts for generating all plots and tables. The model simulations are run over 3000 steps, each representing one working day. Each of the nine scenarios we analyse, as described in the following section, is run 50 times and all variables are averaged over these replicates. In all scenarios, the company has a fixed workforce of 100 employees positioned on a 10*x*10 grid. However, the spatial component plays a role only when a local environment for social norms is activated, see S2 Section in S1 File. The value-types are evenly distributed such that there are 25 employees of each type (C, O, SE, ST). A working day (one simulation step) is comprised of 10 working hours which is the total time endowment $\tau$ to be allocated over the three available uses. The amount of working hours is arbitrary. We assume an intermediate degree of task interdependence, $\kappa = 0.5$, and a norm adjustment of 10%, $h = 0.1$. Different probability distributions of agent types, along with different levels of task interdependence and influence of norms on behaviour have been tested and are included in S3 Section in S1 File. A complete list of agent and model parameters is given in S1.1 and S1.2 Tables in S1 File.

## 3 Results

Our main research question is how incentives set by different remuneration systems affect shirking and cooperation and hence also output in different organisational cultures. In order to answer this question, we first analyse the effects of culture and of the remuneration systems in isolation, before looking at their joint effects. In total, we compare nine different scenarios as shown in Table 3.

The cases in which the management style is neutral or there is no PFP scheme serve as benchmark cases for the later analysis. A "neutral management style" means that there are not autonomy effects as the ones described in Section 2.3.3. In other words, the distributions of the deviations from the shirking norm ($s_t^*$) are centered around the norm for all types.

Fig 8 contains the main results of our simulations. It shows the output that the employees produced with their chosen time allocation as a percentage of the optimal group output (*OGO*). The *OGO* is the maximum output a company can achieve, if all employees choose the identical optimal allocation of their time budget on individual task and collaboration. It is given by the parameter and $\tau$ and $\kappa$ and the Cobb-Douglas function: $OGO = (\tau * (1 - \kappa))^{1-\kappa} * (\tau * \kappa)^{\kappa}$.

We start the discussion by looking at the baseline cases first. The bold black line shows the absolute baseline case with a neutral management style and without PFP incentives schemes. In that case, realised output increases very slowly from 64.5% in period 1 to 67% in period 3000.

The management style has a large impact on realised output. Output grows steadily and reaches the highest level of all cases (94%) in the *Trusting* scenario. In stark contrast, in the *Controlling* one output falls to 0% in the long-run. Such a long-run equilibrium is highly unrealistic, since the management would certainly intervene to counteract a fall in realised output. However, introducing endogenous management decisions is out of the scope of this paper and is left for future research.

**Table 3. Management style, reward schemes and scenarios.**

| | | Management Style | | |
|---|---|---|---|---|
| | | **Neutral** | **Trust** | **Control** |
| PFP Schemes | None | *Base* | *Trusting* | *Controlling* |
| | Group | *Cooperative* | *Trustcoop* | *Contrcoop* |
| | Individual | *Competitive* | *Trustcomp* | *Contrcomp* |

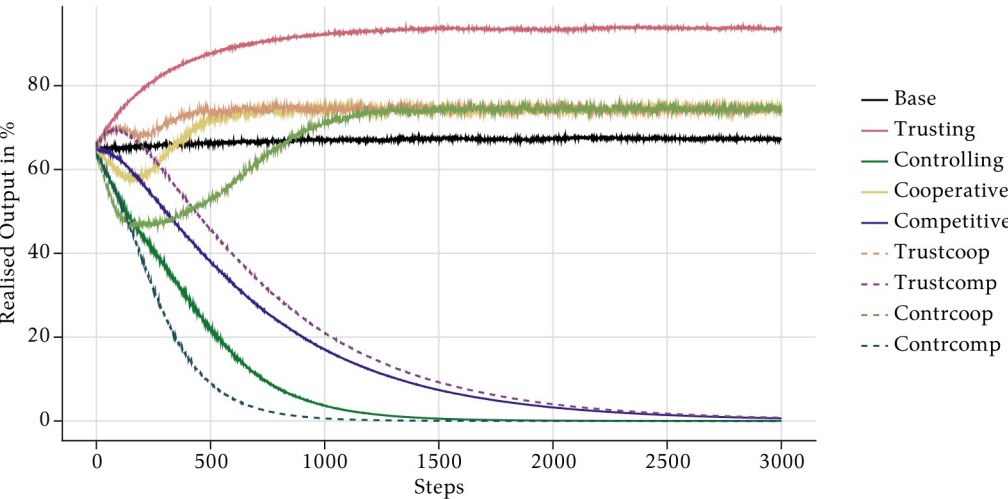

**Fig 8. Aggregate realised output per scenario.**

PFP schemes also cause different evolutions of output. The group bonuses in the *Cooperative* scenario first lead to a decline in output for about 170 periods. After that, output increases again and reaches 74% in period 3000 which is approximately the same level as in the absolute baseline case. In the *Competitive* scenario, output appears stable in the very short-run and then declines at an accelerating rate to about 0% in the last simulation period. However, this convergence towards the long-run equilibrium happens less rapidly than in the *Controlling* scenario.

The outcomes of the scenarios that combine a certain management style with PFP reward schemes are mixtures of the underlying baseline scenarios. If reward schemes are used in a trusting environment, output first increases irrespective of the type of rewards. However, the positive effects of rewards disappear after about 100 periods and output starts falling under both PFP schemes. In a *Trustcoop* scenario the evolution is similar to the *Cooperative* one, which means that the decline in output stops after a while and output starts increasing again until it reaches slightly more than 70% at the end of the simulation period. As in the *Competitive* case, output permanently falls after the initial increase in a *Trustcomp* scenario. The output path in *Contrcoop* mimics the path in *Cooperative*, but at a lower level in the short-run. Finally, the combination of a controlling management style and individual rewards in *Contrcomp* leads to the lowest performance of all cases. The path of output is almost identical to the one in *Controlling* in the first 100 periods, but then falls even faster.

In the remainder of the section, we analyse the reasons of these results.

## 3.1 Absolute baseline scenario

In order to understand the mechanisms of the model, we start with an analysis of the absolute baseline scenario in which the management style is neutral and no PFP scheme is implemented. Fig 9 shows that average individual working time over all employees steadily increases, whereas cooperation time and shirking time decrease. This explains the steady but slow increase in aggregate realised output in Fig 8. Individual working time is the residual determined by the shirking time and the time spent on cooperation. If both go down, there is more time left for the individual tasks. Less cooperation lowers the output of each individual, but this effect is compensated by less unproductive shirking time and more individual production.

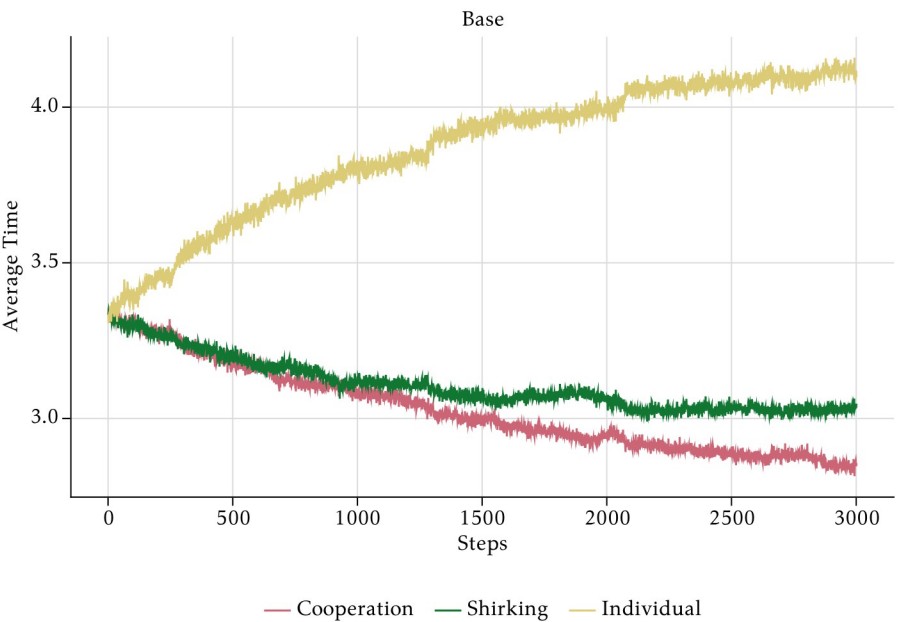

**Fig 9. Aggregate average time—Absolute baseline.**

Fig 10 displays the different behaviours of the four value-types. The upper left panel reveals the ranking of the types in terms of realised output: SE-type agents produce most output, followed by C-agents, O-agents and lastly ST-agents. The other panels show why this is the case. Since the level of shirking is identical for all types (lower right panel), the reason for the different output levels is the different amount of time spent on cooperating with others. In line with the assumption about cooperation (see Fig 5), ST-agents cooperate most and SE-agents least on average, while the other two types are in between. Since SE-agents spend most time on

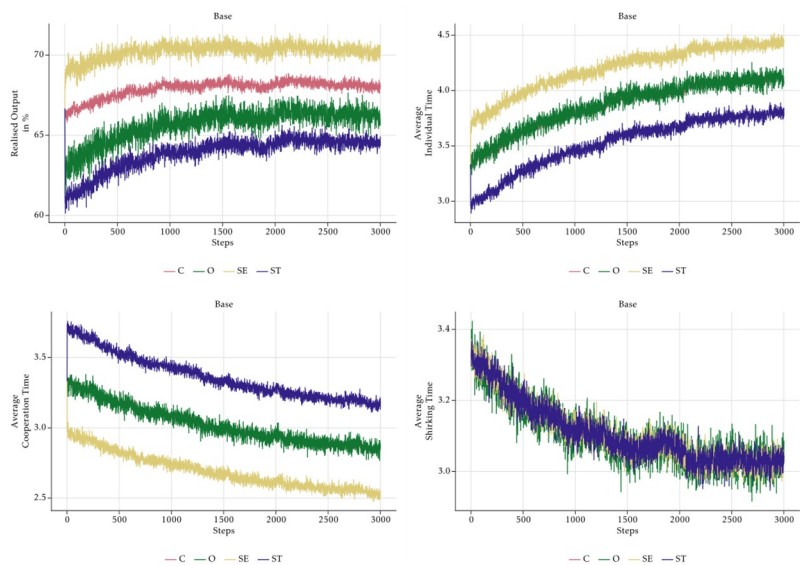

**Fig 10. Realised output, average individual, cooperation and shirking time per value-types—Absolute baseline.**

individual tasks, it is clear that they can produce most. By cooperating, ST-agents increase the output of the other agents, which lowers the time they spend on their own tasks and hence their own output. It is not immediately obvious why C-agents produce more than O-agents, although both spend the same time on their individual tasks on average. The reason is that the variation in O-agents' behaviour is larger than the variation in the behaviour of C-agents. In every period, some O-agents choose an unfavourable combination of high shirking and high cooperation, which leads to low individual task time. Although this is compensated by others, who choose low shirking and low cooperation, the effect is not symmetric due to the decreasing marginal product of individual task time. Higher variation hence leads to lower group output despite the roughly equal average behaviour of the two groups. We checked this hypothesis by lowering the variation in O-agents' behaviour. If the variation gets closer to the one of C-agents, the O-agents' realised output approaches the one of C-agents.

Note the effect of the social norm for cooperation. For all types, cooperation time goes down in lockstep, because the behaviour of all agents is anchored by the social norm. Shirking time decreases, too, as a result of a change in the social norm. The somewhat surprising decreasing trend of the social norms is also caused by the behaviour of the O-agents. If we impose lower variability on O-agents' behaviour, the trend disappears as shown in Fig 11.

## 3.2 Effects of the management style

Next, we isolate the effects of the management style and assume that there is no PFP scheme. If the management style is *Trusting*, cooperation time is constant at 3 hours on average, whereas shirking time tends towards zero over time (Fig 12). As a consequence, individual working time is the mirror image of shirking time and grows during the course of the simulation period. With a *Controlling* management style, the outcomes look rather different. Shirking increases, reaching its maximum at the end of the simulation periods, and cooperation converges towards zero, first slowly and later at a higher rate. The high level of shirking is unrealistic. In reality, the management would not only monitor the employees but also take measures to prevent them from

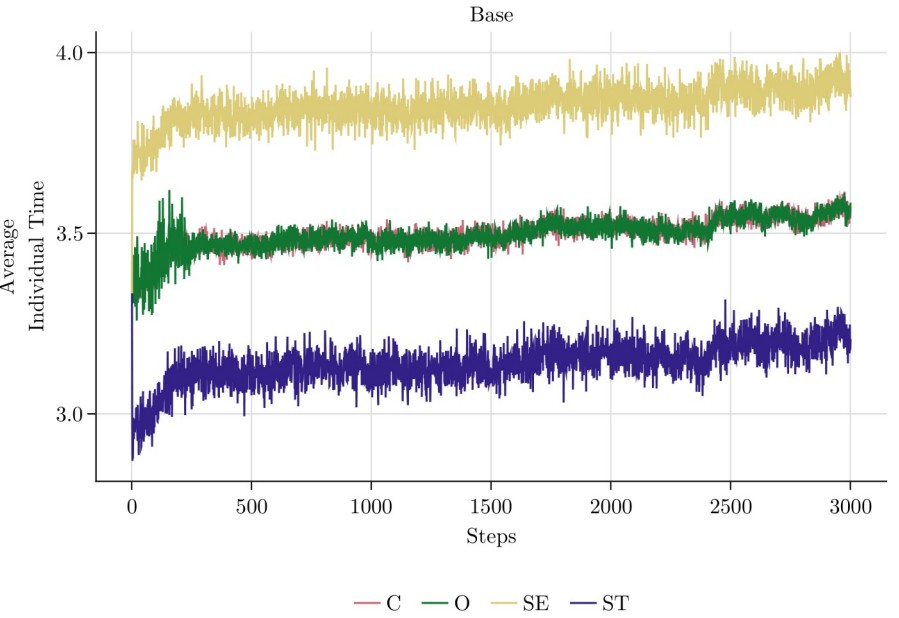

**Fig 11. Effect of lower variability on O-agents' behaviour.**

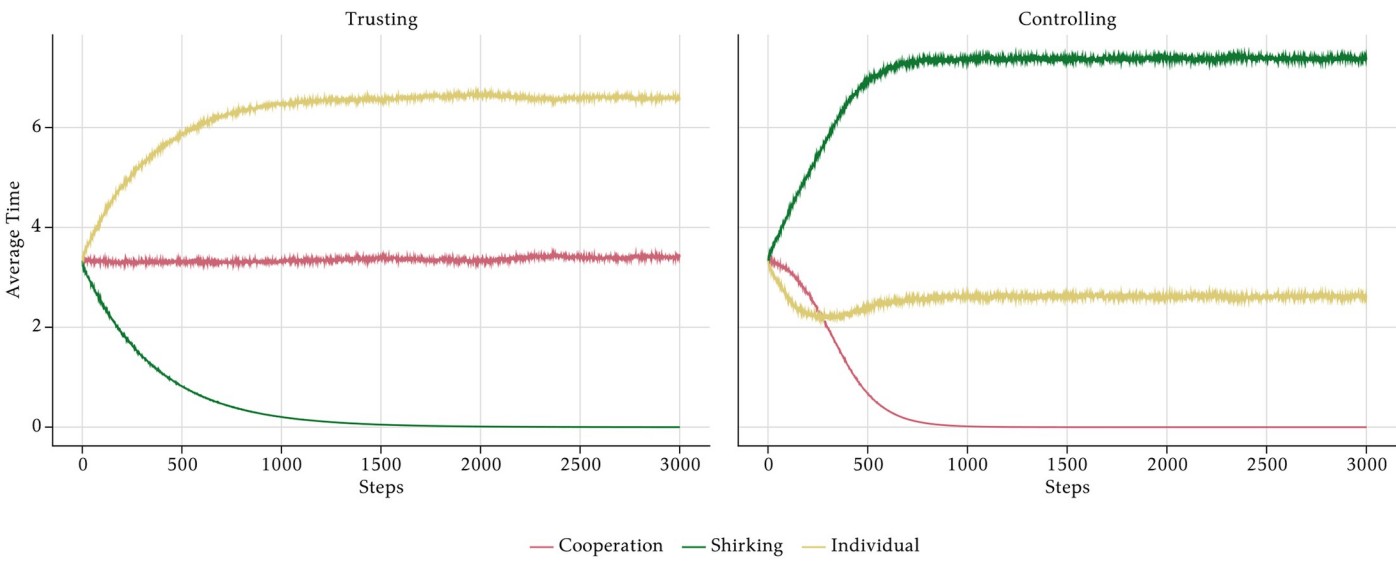

**Fig 12. Aggregate average time—No incentives.**

shirking too much. We neglect such measures here, because we want to isolate the pure motivational effects. The PFP schemes are a motivational device aimed at the prevention of shirking. The dynamics of shirking and cooperation lead to a slightly U-shaped evolution of individual working time which at first decreases, because the increase in shirking is stronger than the decrease in cooperation time. It then stabilises slightly above 2 hours, because the absolute changes of shirking time and cooperation time are equal. Only at around period 500, the increase in shirking slows down, leading to a weak increase in individual working time again.

Fig 12 explains, why output rises if the management style is *Trusting* and shrinks if it is *Controlling*. In our model, control leads to significant shirking, whereas shirking disappears when the management trusts the employees. This effect is driven by the behaviour of the O-agents whose intrinsic motivation is crowded out by a controlling management style.

The bottom panel of Fig 13 shows that O-agents shirk less than the other agents in a trusting culture, but more in a controlling culture. Note that this is the only effect that is directly built into the model. Note also that especially in the trusting culture, the difference in the shirking behaviour of the four types is rather small. Nevertheless, the high intrinsic motivation of the O-agents in the trusting culture is sufficient to drive the shirking behaviour of all types down to almost zero. Analogously, the demotivation of O-agents caused by the controlling management style leads to more shirking of all other agents, too. This is the effect of the social norm. Small systematic deviations of one agent-type are enough to influence the norm and hence affect the behaviour of all employees in the long run.

A surprising effect is the significant drop of cooperation time in the *Controlling* scenario. This effect is the result of a vicious cycle emergent from social norms and agents' time constraint. In a *Controlling* scenario, O-agents shirk more than the norm. Some O-agents shirk so much that their remaining time budget does not suffice to cooperate according to the cooperation norm ($c_t^*$), even if they intend to. If they intend to cooperate more than their remaining time budget, they cannot and are constrained to invest all of their remaining time on cooperation and nothing on their individual tasks. The systematic negative deviation of some O-agents who shirk a lot leads to a decline of the overall cooperation norm. This also reduces SE- and ST-agents' cooperation time as a consequence of the effects of the norm on their degree of

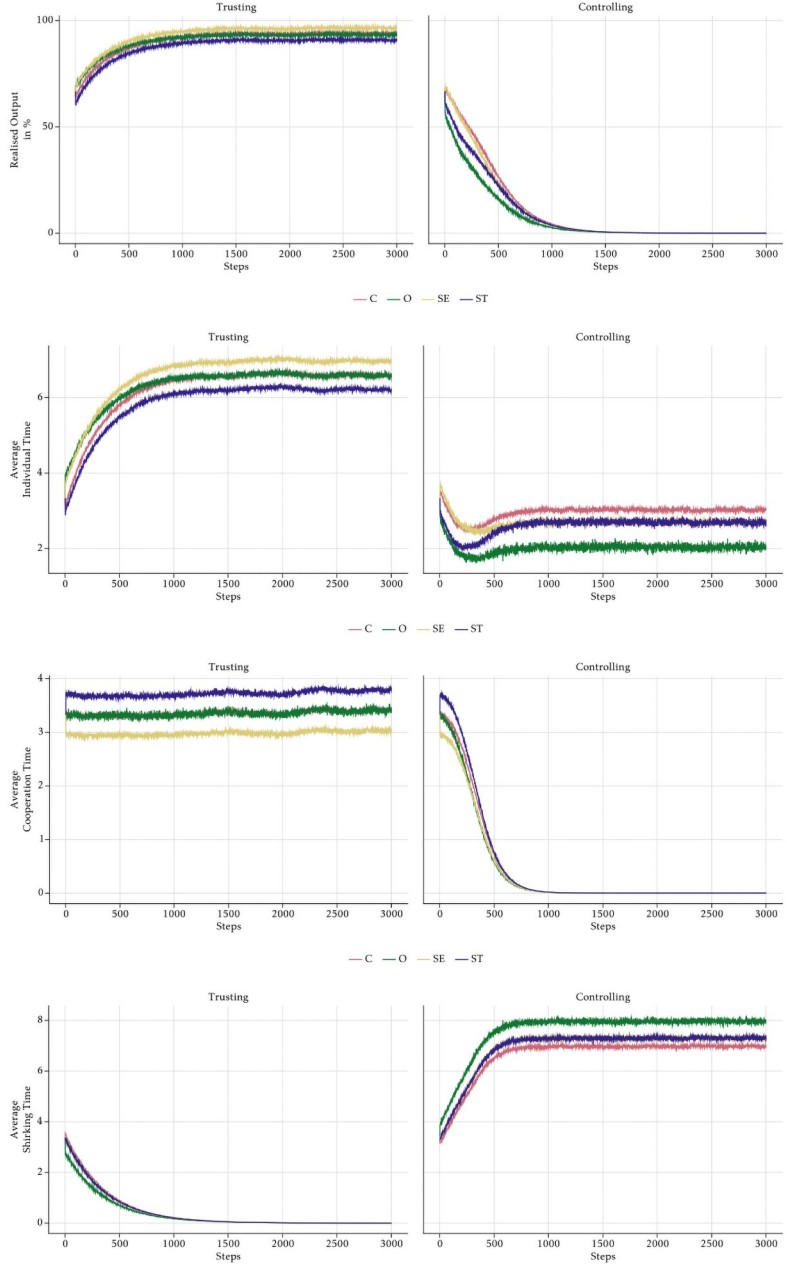

**Fig 13. Realised output, average individual, cooperation and shirking time per value-types—No incentives.**

cooperativeness ($\gamma$). Therefore, because of time constraints, even a small fraction of O-agents, the ones having the highest probability to deviate from social norms, will have a significant influence on cooperative outcomes and on how norms develop within a company.

### 3.3 Effects of the reward scheme

In order to isolate the effects of the PFP reward schemes, we assume that the management style is neutral and has no effect on shirking. Fig 14 shows the effects of the two PFP schemes

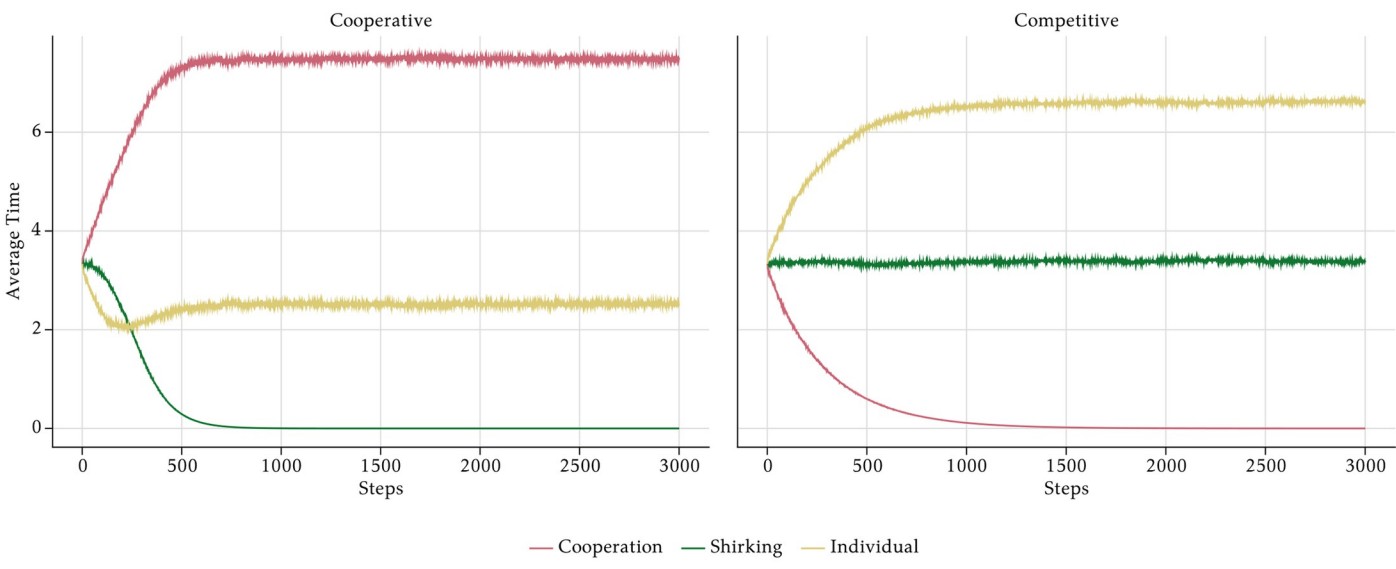

**Fig 14. Aggregate average time—No monitoring.**

on the aggregate time use of the employees. In the *Cooperative* PFP setting with group bonuses, employees' output is mainly driven by rising levels of cooperation and declining shirking, which converges to zero. On the contrary, the *Competitive* scenario with individual bonuses is characterised by rising individual production time, which is the result of a rather stable shirking time and declining cooperation.

Note that the shape of the graphs in Fig 14 and in Fig 12 are analogous. In the *Cooperative* scenario shirking evolves like cooperation in the *Controlling* scenarios. As shown in Fig 15, the explanation is similar. Group bonuses by assumption have no direct effect on shirking, but only affect cooperation. They make the already cooperative ST-agents even more cooperative. Despite the lack of a direct effect of the group bonuses, shirking declines due to a virtuous cycle effect that is similar to the vicious cycle leading to the breakdown of cooperation in the *Controlling* scenario. Again, the time constraint is the cause, but now because of ST-agents who want to cooperate so much that they are forced to shirk less than suggested by the social norm. The strong reduction of shirking of all employees is the reason why the overall output in the scenario is rather high at the end of the simulation. One might doubt that the overall benefit of group bonuses is positive. Although it is clearly desirable from the perspective of the management that agents reduce their shirking behaviour, this has a very strong effect on cooperation because it reinforces the cooperative behaviour of the ST-agents, which is already higher than the average. This increases the cooperation of all agents to an inefficient level. With $\kappa = 0.5$ all agents should allocate their time evenly on individual tasks and cooperation, but due to the group bonuses there is clearly too much cooperation by all agents, but in particular by the agents of the ST-type. Accordingly, the average output of the ST-agents is rather low and only about half of the output of the SE- and the C-agents. The evolution in the *Competitive* scenario is straightforward to explain. Individual bonuses induce ST-agents to cooperate less than the other types, which impairs the cooperation norm. Since shirking is constant, individual working time goes up for all agents, but mostly for the ones of the SE-type. The overall effect of individual bonuses on output is negative, because they crowd out cooperation without affecting shirking. Under the chosen parameter of task interdependence, cooperation is important for the production of output. Due to declining marginal products of individual

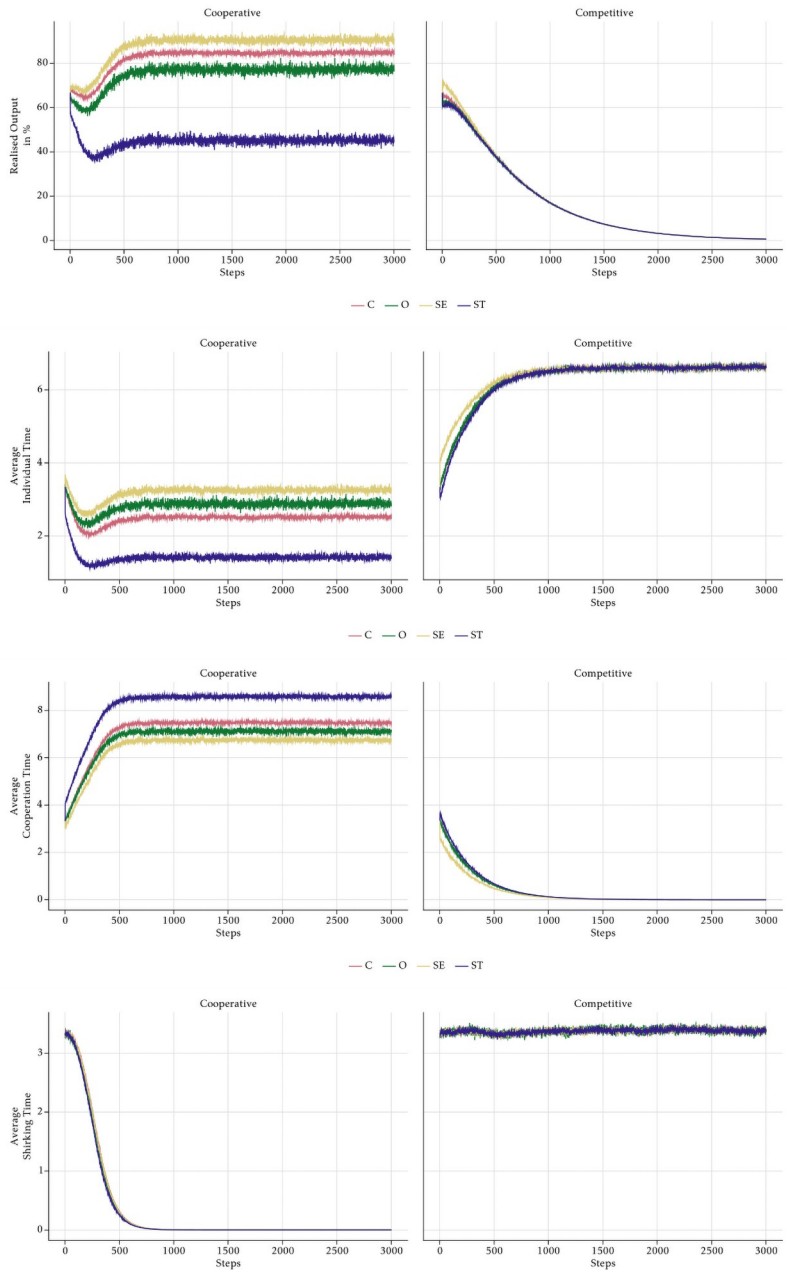

**Fig 15. Realised output, average individual, cooperation and shirking time per value-types—No monitoring.**

working time and cooperation, higher individual effort cannot compensate the loss of cooperation such that output goes down.

## 3.4 Combined effects of management style and rewards scheme

Against the backdrop of the separate effects of the management style and the reward scheme, it is easy to see why *Trustcoop* leads to the highest output of all combinations and *Contrcomp*

generates the lowest performance. The output effects result from the time allocations shown in Fig 16.

Both the trusting management style and the cooperative reward scheme lead to a significant reduction of shirking. While the effect of trusting on cooperation is moderate, the cooperation effect of group bonuses is strong. Accordingly, in *Trustcoop* the total cooperation effect is large, and so is the effect on the suppression of shirking. *Contrcomp* is an unfavourable combination, because both a controlling management style and competitive rewards crowd out cooperation. At the same time, controlling leads to high shirking due to the demotivation of O-agents. Therefore, the combined effect is high shirking and low cooperation, which leads to low output. *Trustcomp* is also a quite successful combination. Although the competitive reward system leads to declining cooperation, output is high, because there is little shirking due to the high motivation of O-agents. In the long run, however, output declines because cooperation is not sufficient. *Contrcoop* initially leads to slightly less output than *Trustcomp*, because shirking remains high for 300 periods, which is due to the demotivation of the O-type agents. Over time, however, the group bonuses increase both cooperation, which is positive, and work

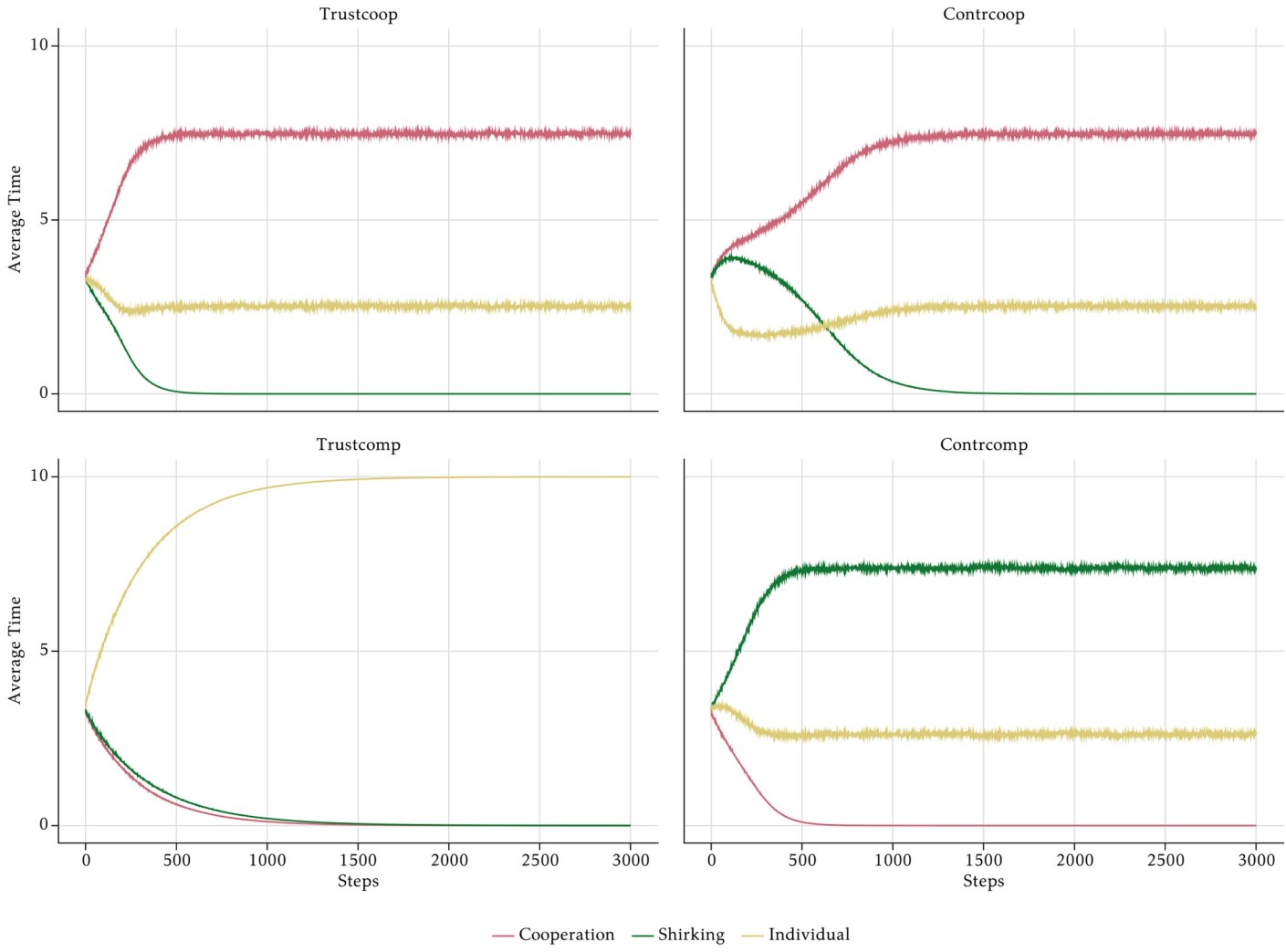

**Fig 16. Aggregate average time—PFP schemes and monitoring strategies.**

against the push from O-agents to more shirking. Both the increasing cooperation and the slowly falling shirking ultimately lead to rising output.

### 3.5 Evaluation of financial incentives

As argued in the introduction, monetary incentives are a widely used instrument in companies to induce desired behaviour of the employees. In our model, however, PFP schemes make little sense. We found that competitive individual bonuses lead to a lower total output compared to the baseline scenario with a neutral management style and a flat wage. In the last simulation period, total output with competitive bonuses is more than 99% lower than with a flat wage. Cooperative group bonuses can increase output compared to the baseline, especially if they are combined with a trusting management style. However, the gain in total output is rather modest and only occurs after about 350 periods if the management style is neutral, and after 850 periods if the management adopts a controlling style (*Contrcoop*). In period 3000, output is just 10 percentage points higher due to cooperative bonuses. This small increase in output is accompanied by a significant increase in total labour costs. In the *Cooperative* and *Trustcoop* scenarios, total financial rewards (base wage + bonuses) over the whole simulation period are about 37% higher than in the baseline scenario.

Despite the implausible convergence to zero of realised output in the long-run, as already pointed out in Section 3, these results seem to suggest that remuneration schemes play a key role in determining the long-run equilibria: Independent of the management style, scenarios with cooperative incentive schemes lead to better performing outcomes in the long run compared to their competitive counterparts.

We showed that both incentive schemes have undesired effects on cooperation, because they reinforce the natural cooperation tendencies of some types of agents. Group bonuses enhance the strong cooperation of ST-agents even more, which leads to inefficiently high social norms of cooperation. Analogously, individual bonuses reduce the already low cooperation time of SE-agents leading to a general erosion of cooperation. In our model, it is more reasonable to increase output by adopting a trusting management style. The change in the management style does not increase labour costs, but increases output by 38% over the whole simulation.

We cannot claim that our results are general and we do not make any statements about the empirical validity of the theoretical findings. As such, confronting our model with empirical evidence is left for future research. Our results demonstrate that under plausible assumptions that are supported by empirical studies, the interaction of value-driven behaviour and social norms can generate unintended consequences of reward schemes. The total long-run effects of financial incentives can be difficult to predict in such a context. As could be seen in Fig 8, there are relevant turning points around step 250 in the time series for the *Cooperative*, *Contrcoop*, and *Trustcoop* scenarios. At this stage, past corporate performance shows decreasing trends, yet will converge towards stable high levels over the course of the following 1000 steps. Therefore, adopting endogenised incentives schemes, management decisions guided by current output and recent developments might lead to undesired results. The short-run decreasing tendency of realised output in a scenario with cooperative incentive schemes might induce the management to change its strategy in favour of competitive schemes or dropping them completely. Keeping the above-mentioned turning points in mind, there is a non-negligible risk of inefficient adjustment of incentive schemes after management observes short-run dips in key metrics. These deliberations need to be scrutinised in future works but currently seem to suggest that the steering effect of incentive schemes is significantly limited by the value composition of the company's workforce (confer also the observations in Test 1 of S3 Section in S1 File).

## 4 Discussion

Our paper is a first step towards the development of a theory of corporate culture that explains how culture affects company performance and how it interacts with elements of organisational structure such as the remuneration system. We conceptualise corporate culture as a mix of the management style chosen by the management and the endogenous descriptive social norms on cooperation and shirking. Our model has been made available as open source code (here) to encourage reproducibility and open collaboration, both necessary to deepen our understanding of heterogeneous agents' decision making processes in corporate settings.

We show that both the management style and the remuneration system influence the endogenous norms with regard to cooperation and shirking and hence have an impact on total output. The "soft" lever of the management concerning the management style has a greater impact on output than the "hard" remuneration lever. In general, financial rewards in the form of PFP bonuses are not recommendable in our model, because they increase output only moderately in some scenarios at considerable cost. The main drawback of PFP bonuses is that they reinforce the natural propensities of cooperative and uncooperative employees too much. The social norm responds to the extreme behaviours of these types and amplifies the effects even further leading to a circular causation. As a consequence, group bonuses lead to excessive cooperation of all agents, while individual bonuses cause a significant decline of overall cooperation, which is not sufficiently compensated by the higher individual efforts.

An important result of our model is that behavioural differences due to values matter a lot for the group outcomes. We assume that a controlling management style demotivates agents who value openness and self-direction highly, which results in higher shirking by these agents. Although other agent-types are not directly affected by the controlling management style, the demotivated agents shift the social norm on shirking upwards and hence induce much more shirking by all employees. Furthermore, there is an unexpected indirect effect of the demotivated O-agents' behaviour on cooperation. If some O-agents shirk so much that they do not have sufficient time left to spend on cooperation, they also drag down the cooperation norm. Group bonuses cause a similar effect via the behaviour of ST-agents, for whom the well-being of others is important. The natural inclination to cooperate with others is strengthened by group bonuses that trigger a normative goal frame and a collective we-orientation. Group bonuses encourage some already very cooperative agents to spend so much time on cooperation that they even shirk less than normal, which finally results in very little shirking by all employees.

Further attempts to overcome some limitations of this model might include (i) endogenous management strategies, (ii) heterogeneous tasks, skills and productivity levels and (iii) adaptive agents' behaviour in a more complex network environment. This is left for future research.

## Supporting information

**S1 File.**
(PDF)

## Author Contributions

**Conceptualization:** Michael Roos.

**Data curation:** Jessica Reale, Frederik Banning.

**Formal analysis:** Jessica Reale, Frederik Banning.

**Funding acquisition:** Michael Roos.

**Methodology:** Jessica Reale, Frederik Banning.

**Project administration:** Michael Roos.

**Software:** Frederik Banning.

**Supervision:** Michael Roos.

**Validation:** Jessica Reale, Frederik Banning.

**Visualization:** Jessica Reale, Frederik Banning.

**Writing – original draft:** Michael Roos.

**Writing – review & editing:** Michael Roos, Jessica Reale, Frederik Banning.

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
