## [Decision Letter · Decision Letter 0]

26 Oct 2021

PONE-D-21-29609A value-based model of job performancePLOS ONE

Dear Dr. Reale,

Thank you for submitting your manuscript to PLOS ONE. After careful consideration, we feel that it has merit but does not fully meet PLOS ONE’s publication criteria as it currently stands. Therefore, we invite you to submit a revised version of the manuscript that addresses the points raised during the review process.

We look forward to receiving your revised manuscript.

Kind regards,

Ali B. Mahmoud, Ph.D.

Academic Editor

PLOS ONE

Journal Requirements:

Reviewers' comments:

Reviewer's Responses to Questions

**Comments to the Author**

1. Is the manuscript technically sound, and do the data support the conclusions?

Reviewer #1: Yes

Reviewer #2: Yes

2. Has the statistical analysis been performed appropriately and rigorously? 

Reviewer #1: Yes

Reviewer #2: Yes

3. Have the authors made all data underlying the findings in their manuscript fully available?

Reviewer #1: Yes

Reviewer #2: Yes

4. Is the manuscript presented in an intelligible fashion and written in standard English?

Reviewer #1: Yes

Reviewer #2: Yes

5. Review Comments to the Author

Reviewer #1: Summary:

Let me first preface my review by saying that I am not an expert in socioeconomic dynamics, but have expertise in agent based simulations.

All in all, I found the paper very interesting. The introduction makes it clear that the paper addresses an existing gap in the literature. Specifically, the authors have identified that most of the existing work does not take into account heterogeneity of agents which is taken explicitly into account in this work. The integration with the established importance of social norms, and how these evolve along with agent behavior also raises the quality of the model. The modelling aspect of the agent heterogeneity is executed very well and is intuitive to grasp. The results of the simulations of the authors show that indeed behavioural differences due to agent heterogeneity matter a lot. An additional impact of the paper is simply the fact that it opens the door to more complex modelling of workplace productivity which can then lead to further research. As such, I believe this paper is worth publishing.

The main improvements I have listed for the manuscript are regarding the clarity with which information is delivered.

In addition the manuscript is quite lengthy. I received 30 pages to review, which I guess after better figure placement would be reduced to 25. Perhaps it is worth considering grouping information more efficiently (e.g., I've noticed that the aspects of heterogeneity are repeated several times, or I've noticed that there are lengthy justifications in the methodology section which could be made shorter, especially given their mentioning in the introduction).

I applaud the authors for making the source code of their work available openly online, which is the only way one can ensure true reproducibility of scientific work. I believe the authors should mention this in Sect. 4, as this makes their model immediatelly accessible to others for further research. Something like "Our model is available as open source code[cite], and could be integrated in decision making protocols based on agent heterogeneity in corporate settings."

Comments:

* The introduction of the paper is well written and gives a good overview of what has been done and the standing view points on the subject. What I found it lacking is that it does not make it crystal clear of what are the specific research questions to be answered by this paper. At 1/3rd of the introduction, the authors point out the novelty of their approach "agents differ in personal values". So the questions are somewhat implicitly assumed as "how does this new approach affect the socioeconomic dynamics of the model". But one should be clearer and explicit in the introduction about the research questions, especially given that the remaining 2/3rds of the introduction continue to review existing literature, but don't come back at what the authors actually want to do.

* Given that one of the premises of the paper is that agents are individuals with different values, I was surprised to read in the start of Sect. 2.1 that agent skills and productivity are all identical. I guess this is a simplified assumption the authors made to reduce the number of parameters of the model. This is okay. One thing I'm wondering is, would the results of the simulations change if the agents did not have identical skills, but rather different skill and productivity levels randomly distributed in an agent population? Can the authors provide some intuitive justification on this, without necessarily re-making lengthy simulations?

* Is the agent coupling all-to-all when computing the average t_{jc}? Please make it clear in the paper that there is no spatial structure in this agent model.

* I am skeptical of the opening paragraph of Sect. 2.2 and also whether it is necessary at all. First of all, Ref. 19, which the entire paragraph is based on, is not a scientific reference published by a scientific publisher, but rather it is a public book. Secondly, it seems that all of this discussion on system1/system2 is not necessary. What is necessary for Sect. 2.2 is the "descriptive norms" assumption, which is already stated in the introduction. In fact, nothing in my perception and understanding of Sect. 2.2 changes once I remove its first paragraph. The only point that system1 is referred to is to justify the inclusion of the stochastic term Δ, which, I think, is an inclusion that needs no real justification. If the first paragraph is indeed important, the authors need to do a better job connecting it with the remaining of the section, and also connecting it with more solid scientific references.

* The statistical sampling of the stochastic component of the model is not explained adequately. This comment ties with my technical comment on the mathematical notation of the paper being hard to navigate. The distribution function f_x is presented, but it is not stated clearly and explicitly which of all the presented variables are actually sampled from f_x. Is it Δs_i and Δc_i? The Δ are presented as the stochastic components of the model, and hence one would assume that these are sampled from a distribution. But that cannot be true, because later we read "In this baseline case, employees would most likely spend their time according to the norm since the expected deviation is zero." which makes me think that f_x in fact samples entire time intervals instead of Δ.

* I think the manuscript can benefit from a figure/sketch that has some kind of arrows or basic shapes that map and illustrate visually the main aspects and main formulas of the model. E.g. one mapping would be "heterogeneous properties" to "actual model parameters and functions changed to reflect this heterogeneity", another box would contain the responsiveness to rewards, etc. For example, one box would be "cooperativeness" which would map to "distribution function for values of Δt_c" (or t_c once the authors clarify their notation of what the distribution functions actually sample).

* Throughout section 2.3 the authors introduce incrementally parameters (such as γ, φ, ...) that alter the main probability distribution function f_x that models all stochastic aspects of the model. However, these parameters are never actually put in the definition of f_x and the reader is left to "derive the new expression" themselves. This also introduces so much unnecessary extra text to the article, where math would do the trick (quite literally) in only a single line. Perhaps the authors want to rethink their approach around how f_x is incorporated in 2.3 and reduce the overall amount of text significantly in favor of clearer math.

* What's the reason to choose 500 days as the total simulation time? Was there any criterion for convergence employed? Do the model dynamics stabilize?

* Sect. 4 is very good. Short, to the point, and highlight all novelties excellently.

Technical comments:

* References are printed with bracket notation, e.g. [7]. However, starting a sentence with such a reference is not appropriate style (e.g., "[7] call for the development...") in second paragraph). The authors should use last name and et al. followed by the refernce. E.g., Chatman et al. [7] call for the development....

* It is easier to the reader if the same word is used for the same notion consistently throughout the text. It seems that e.g., "company" and "firm" are used interchangeably. Eliminating different words used for the same concept will make the text easier to read and follow.

* Equations (4,5) etc. started to become difficult to follow because of the way the time indexing works. At the start of Sect. 2.1 the authors say that "for notational convenience they do not use a time index", which is a decision I certainly do not agree with. Then, in (4) a secondary subscript (-1) is used for the time index of the previous step, even though the variables of the current time step have no time index. Furthermore, the sudden appearance of an asterisk * for the t variables makes things even more confusing. It is not explained in text what the asterisk represents until it is too late. Only when one reaches equation 7 it becomes clear that the asterisks denote some mean value. I think the mathematical exposition of the formulas can be improved for more clarity. I also wonder, is it indeed better to have t with subscripts for all task times, instead of having three variables, p c and s instead?

* Why is f_x written f_x instead of f(x), given that it is a function of x?

* After Fig. 1: "hence the mode is equal to the social norm" please say the "mode of the distribution".

* "Factor analyses show that the basic values can be aggregated along two dimensions." Needs a citation.

* "deviations from the shirking norm and the cooperation norm" I believe throughout the text such references can be improved by explicitly referencing the associated symbol. If I have understood the methodology, the cooperation norm is just $t_c^*$. Hence, this symbol should be added immediately after "cooperation norm".

Reviewer #2: The manuscript aims to find the effect of interactions between monetary incentives, intrinsic motivations, and social norms on work productivity. The novelty of the manuscript is in modeling a population of employees with heterogenous intrinsic motivations. The heterogeneity of employees proves to be the most important factor in determining the overall productivity of the employees in different reward and management schemes. Since social norm is the parameter that ultimately determines the employee behavior in the model, the subject of the study can be thought of as the evolution of social norm. To tackle the problem, the authors use a bottom-up approach with agent-based modeling, which is a suitable method for such complex problems.

The manuscript is well written. The theory behind every decision for modeling is explained simply enough that I could easily follow it as someone with no formal background in management. The model itself is clearly laid out and the results are explained in detail. I also appreciate the sensitivity analysis. I recommend the paper for published given the issues below are addressed.

1. The choice of the four employee types (ST, SE, C, and O) needs further justification. The types are on two independent dimensions (ST-SE, and C-O). Thus, an employee can be open to change/conservative AND self-transendent/self-enhancing. Therefore, the more intuitive types would be ST-O, ST-C, SE-O, and SE-C. This would change values in Table 1 and density functions, potentially simplifying the model because there will be no intermediate distribution curves anymore.

2. Additionally, the choice for the length of simulations (500 days) is not discussed. Plots show that populations do not reach equilibrium until the end of the simulations (although they do not seem to be changing course either).

3. O-agents are important drivers of the evolution of social norms because of their larger variance in deviating from norm compared to other employees. Figure 9 proves this effect. At first, it sounds counter intuitive that the positive and negative deviations from the norm do not cancel each other out. Since this is such an important determinant of the model behavior, it deserves a more detailed explanation than the single sentence given on page 20.

4. Figure quality is too poor and I had difficulty reading the labels. Using a vector format for saving images (e.g. PDF) should solve this.

5. The Discussion benefits from expanding on the limitations of the study. For example, although the study incorporates heterogenous agents instead of a single type, there are other parameters kept fixed that could affect the results. Tasks are not homogenous either, some requiring more collaboration and some less. What would be the effect of a mixed management style depending on the amount of collaboration needed for a task? I suppose in such scenarios, TrustComp would be a winning strategy.

6. PLOS authors have the option to publish the peer review history of their article (what does this mean?). If published, this will include your full peer review and any attached files.

Reviewer #1: **Yes: **George Datseris

Reviewer #2: No

---

## [Author Response · Author response to Decision Letter 0]

7 Dec 2021

Please see the attached pdf "Response to Reviewers".

---

## [Decision Letter · Decision Letter 1]

15 Dec 2021

PONE-D-21-29609R1A value-based model of job performancePLOS ONE

Dear Dr. Reale,

Thank you for submitting your manuscript to PLOS ONE. After careful consideration, I can accept your article after undergoing a few minor corrections suggested by Reviewer #1. Therefore, I mark this decision as a conditional acceptance, and I invite you to submit a revised version of the manuscript that addresses the points raised during the review process.

We look forward to receiving your revised manuscript.

Kind regards,

Ali B. Mahmoud, Ph.D.

Academic Editor

PLOS ONE

Journal Requirements:

Reviewers' comments:

Reviewer's Responses to Questions

**Comments to the Author**

1. If the authors have adequately addressed your comments raised in a previous round of review and you feel that this manuscript is now acceptable for publication, you may indicate that here to bypass the “Comments to the Author” section, enter your conflict of interest statement in the “Confidential to Editor” section, and submit your "Accept" recommendation.

Reviewer #1: All comments have been addressed

Reviewer #2: All comments have been addressed

2. Is the manuscript technically sound, and do the data support the conclusions?

Reviewer #1: Yes

Reviewer #2: Yes

3. Has the statistical analysis been performed appropriately and rigorously? 

Reviewer #1: Yes

Reviewer #2: Yes

4. Have the authors made all data underlying the findings in their manuscript fully available?

Reviewer #1: Yes

Reviewer #2: Yes

5. Is the manuscript presented in an intelligible fashion and written in standard English?

Reviewer #1: Yes

Reviewer #2: Yes

6. Review Comments to the Author

Reviewer #1: The authors have addressed all of my comments satisfactorily. The paper is now in a great shape, and I recommend publication straight out. Figure 1 is just incredible!!!

I only point out some typos and minor comments:

* In equation 2 the subscript of \\bar{c} on the left-hand-side of the equation should be {i, t} instead of {j, t}. And this is also true for equation 1. Given that equation 2 defines \\bar{c} so that the right-hand-side is NOT a function of j, as j is reduced via summation, it cannot be that the left-hand-side is a function of j either.

* Equation (4) must end with a period "."

* Page 7 first paragraph ends with: ", and the average of all agents’ behaviour in the previous period:" So it ends with colon, but nothing follows the colon; no equation but rather a new paragraph.

* Equation (5) must end with a period "."

* When introducing the triangular distribution x_t ~ T(a,b,m) please say that m is the mean. Furthermore, please write "the natural bound a = m - m = 0" (i.e., explicitly add " = 0")

Reviewer #2: (No Response)

7. PLOS authors have the option to publish the peer review history of their article (what does this mean?). If published, this will include your full peer review and any attached files.

Reviewer #1: **Yes: **George Datseris

Reviewer #2: No

---

## [Author Response · Author response to Decision Letter 1]

22 Dec 2021

Please see the attached pdf "Response to Reviewers".

---

## [Decision Letter · Decision Letter 2]

23 Dec 2021

A value-based model of job performance

PONE-D-21-29609R2

Dear Dr. Reale,

We’re pleased to inform you that your manuscript has been judged scientifically suitable for publication and will be formally accepted for publication once it meets all outstanding technical requirements.

Kind regards,

Ali B. Mahmoud, Ph.D.

Academic Editor

PLOS ONE

Additional Editor Comments (optional):

Reviewers' comments:

Reviewer's Responses to Questions

**Comments to the Author**

1. If the authors have adequately addressed your comments raised in a previous round of review and you feel that this manuscript is now acceptable for publication, you may indicate that here to bypass the “Comments to the Author” section, enter your conflict of interest statement in the “Confidential to Editor” section, and submit your "Accept" recommendation.

Reviewer #1: All comments have been addressed

2. Is the manuscript technically sound, and do the data support the conclusions?

Reviewer #1: Yes

3. Has the statistical analysis been performed appropriately and rigorously? 

Reviewer #1: Yes

4. Have the authors made all data underlying the findings in their manuscript fully available?

Reviewer #1: Yes

5. Is the manuscript presented in an intelligible fashion and written in standard English?

Reviewer #1: Yes

6. Review Comments to the Author

Reviewer #1: (No Response)

7. PLOS authors have the option to publish the peer review history of their article (what does this mean?). If published, this will include your full peer review and any attached files.

Reviewer #1: **Yes: **George Datseris

---

## [Editor Report · Acceptance letter]

13 Jan 2022

PONE-D-21-29609R2 

A value-based model of job performance 

Dear Dr. Reale:

I'm pleased to inform you that your manuscript has been deemed suitable for publication in PLOS ONE. Congratulations! Your manuscript is now with our production department. 

Kind regards, 

on behalf of

Dr. Ali B. Mahmoud 

Academic Editor

PLOS ONE